# Outdoor Walking Classification Based on Inertial Measurement Unit and Foot Pressure Sensor Data

**DOI:** 10.3390/s26010232

**Published:** 2025-12-30

**Authors:** Oussama Jlassi, Jill Emmerzaal, Gabriella Vinco, Frederic Garcia, Christophe Ley, Bernd Grimm, Philippe C. Dixon

**Affiliations:** 1Department of Kinesiology and Physical Education, McGill University, Montreal, QC H2W 1S4, Canada; oussama.jlassi@mcgill.ca (O.J.); jill.emmerzaal@mcgill.ca (J.E.); 2Department of Mathematics, University of Luxembourg, 4365 Esch-sur-Alzette, Luxembourg; gabriella.vinco@uni.lu (G.V.); christophe.ley@uni.lu (C.L.); 3Department of Precision Health, Luxembourg Institute of Health, 1445 Luxembourg, Luxembourg; frederic.garcia@lih.lu (F.G.); bernd.grimm@lih.lu (B.G.)

**Keywords:** gait analysis, walking condition classification, wearable sensors, inertial measurement units, pressure insoles, digital mobility outcomes, machine learning, deep learning

## Abstract

(1) Background: Navigating surfaces during walking can alter gait patterns. This study aims to develop tools for automatic walking condition classification using inertial measurement unit (IMU) and foot pressure sensors. We compared sensor modalities (IMUs on lower-limbs, IMUs on feet, IMUs on the pelvis, pressure insoles, and IMUs on the feet or pelvis combined with pressure insoles) and evaluated whether gait cycle segmentation improves performance compared to a sliding window. (2) Methods: Twenty participants performed flat, stairs up, stairs down, slope up, and slope down walking trials while fitted with IMUs and pressure insoles. Machine learning (ML; Extreme Gradient Boosting) and deep learning (DL; Convolutional Neural Network + Long Short-Term Memory) models were trained to classify these conditions. (3) Results: Overall, a DL model using lower-limb IMUs processed with gait segmentation performed the best (F1=0.89). Models trained with IMUs outperformed those trained on pressure insoles (p<0.01). Combining sensor modalities and gait segmentation improved performance for ML models (p<0.01). The best minimal model was a DL model trained on IMU pelvis + pressure insole data using sliding window segmentation (F1=0.83). (4) Conclusions: IMUs provide the most discriminative features for automatic walking condition classification. Combining sensor modalities may be helpful for some model architectures. DL models perform well without gait segmentation, making them independent of gait event identification algorithms.

## 1. Introduction

Walking in the natural and built outdoor environment is a common task for most individuals that requires navigating various surface topologies such as stairs, ramps, or irregular surfaces. Numerous studies have already shown that surface topology alters biomechanical gait patterns [1,2,3,4,5,6,7,8]. These adaptations can be challenging for aging populations and individuals with gait or mobility issues [2,8]. In fact, uneven surfaces have been identified as a major factor in outdoor falls among middle-aged and older adults [9]. As such, incorporating outdoor environments in biomechanical analysis would fully capture the demands of walking. Accordingly, valid and easy-to-use algorithms to classify walking conditions for remote monitoring of real-life behavior are required.

Previous studies have explored the use of various sensor modalities for walking surface classification using machine learning (ML) and/or deep learning (DL) approaches. In total, we found 17 studies that used wearable sensors to classify surface topologies with variable success (Table A1). Most studies used only IMUs (n = 13), while four studies used a multimodal sensor approach. Multiple studies have implemented algorithms capable of classifying outdoor walking surfaces with very high accuracy (>0.95) [10,11,12,13,14,15,16,17,18]; however, results were based on model training without stratification by subject (record-wise or random split). Within model training, using a random train/test split without stratification by subject may result in data from the same participant in both the training and test set. This is a clear case of non-independence between training and test samples, resulting in a form of data leakage that can lead to overconfident model performance expectations [19]. In fact, several studies have shown that using a random split instead of a subject-stratified split greatly improves the classification results, leading to overfitted models for this particular problem statement [10,14,20]. Therefore, it is advantageous to develop models that are trained subject-wise for improved generalizability of model performance to unseen participants.

Many studies have also achieved near perfect classification via the use of numerous sensors (>3 in [10,11,15,17,18,21,22]); however, such set-ups may result in compliance issues during real-world monitoring. As such, minimally obtrusive sensors such as pressure insoles could be a viable alternative. Recent work has shown that vertical ground reaction forces and center of pressure captured with pressure insoles are influenced by surface topology [23]; however, only limited research has implemented pressure insoles for surface classification. In fact, we found only a single pilot study that incorporated pressure insoles in a multimodal classification approach [18]. They found near perfect classification accuracy, but recruited only five participants who walked on indoor surfaces, and implemented random splitting [18]. Therefore, it remains unclear whether algorithms based on pressure insoles could provide strong walking condition prediction capabilities. Moreover, all previous surface classification models have relied on highly segmented walking bouts on the different surfaces collected using short individual trials, suggesting that the ecological validity of models has yet to be established for unconstrained walking.

Finally, the effect of data segmentation on model performance is largely unknown. Previous work has used either segmentation on gait cycles (cf. [10]) or no segmentation at all (cf. [11]). As walking condition can influence biomechanics, and consequently the signal representation within different steps, it is crucial to explore whether data segmentation methods matter. There are two common methods: the traditional biomechanical gait cycle segmentation and the more data-driven sliding window approach. Segmentation based on gait cycles requires domain knowledge and identification of gait events, whereas fixed window segmentation requires no domain knowledge and makes the resulting models independent of gait event identification algorithms.

Therefore, the goal of this study is to develop and validate tools for automatic walking condition classification using IMU sensor and pressure insole data. The primary aim is to compare the performance of different sensor modalities (IMU, pressure insoles, and combined) in identifying the walking condition. The secondary aim is to determine whether gait cycle segmentation improves model performance compared to fixed window approaches. We hypothesize that: (1) IMU sensors will provide more discriminative information than pressure insoles, with models combining IMU and pressure insole enhancing performance; and that (2) gait cycle segmentation will outperform sliding window segmentation. By addressing these research aims, this study contributes to enhancing the ecological validity of ML and DL models in classifying surface topology, thereby aiding biomechanical analysis to be unconstrained during outdoor walking.

## 2. Materials and Methods

### 2.1. Data Acquisition and Segmentation

We leveraged a pre-existing dataset by Losing & Hasenjäger (2022) [24] in this study. Briefly, the dataset represents a motion capture experiment that recorded data using an Xsens motion capture suit consisting of 17 IMU sensors [25] and IEE ActiSense Smart Footwear Sensor pressure insoles [26] as participants walked across real-world environments. The data were collected from 20 participants (five females, ages 18–69 years) walking three courses of approximately 500 m (Courses A, B, and C) where they encountered various walking conditions. This dataset comprises unconstrained outdoor walking that includes common surface transitions and interaction with other people (e.g., side-stepping). These natural perturbations make it an excellent choice for training surface classification models with realistic performance expectations. Data were provided at the downsampled rate of 60 Hz. For additional details, see the original publication (cf. [24]).

### 2.2. Data Preparation

For this study, we reduced the dataset from seven classes to the five classes that were annotated in the database: flat walking, walking up stairs, walking down stairs, walking up slopes, and walking down slopes. The pavement up and pavement down classes were removed, as they only represent transient transitions on and off curbs spanning a single step. We included only the tri-axial accelerometer and gyroscope signals from the IMU sensors, excluding the magnetometer due to its susceptibility to magnetic interference. As the dataset expresses the IMU data in the global coordinate frame, we reoriented the coordinate system to each segment’s local coordinate system following the International Society of Biomechanics (ISB) recommendations [27] to ensure that changes in heading would not affect model performance. This transformation was implemented using custom Python code (https://github.com/jillemmerzaal/heading-direction, accessed on 8 April 2025).

#### 2.2.1. Sensor Configurations

We used two IMU sensor configurations. First, we used a seven-sensor “lower limb” configuration with sensors on the pelvis, upper legs, lower legs, and feet (Figure 1). This lower limb model has performed well with predicting surface topology in previous work (cf. [11]) and is referred to herein as the “established model”. To investigate minimal IMU configurations, we evaluated setups comprising IMUs mounted on the feet only (left and right; two sensors) and on the pelvis only (one sensor). The pelvis-only and foot-only configurations were used to assess performance under minimal instrumentation. Moreover, the foot-only IMU configuration enabled fair comparison with foot pressure-based models. The pressure insoles comprised eight sensors: arch, hallux, left heel (heel L), right heel (heel R), first metarsal (met 1), third metatarsal (met 3), fifth metatarsal (met 5), and toes. We included the pressure data normalized to each participant’s body mass, available from the public dataset, as inputs to the pressure-based models. Overall, for both ML and DL approaches, we evaluated six models: (1) IMU lower limbs (seven IMUs), (2) IMU feet (two IMUs), (3) IMU pelvis (one IMU), (4) pressure feet, (5) IMU feet + pressure feet, and (6) IMU pelvis + pressure feet.

#### 2.2.2. Data Segmentation

We implemented two distinct data segmentation strategies: gait-based segmentation and sliding window segmentation. Gait-based segmentation relied on gait event annotations derived from the pressure insole data provided in the dataset, where instances of foot–ground contact were identified [24]. Using these annotations, individual gait cycles were segmented and time-normalized to 101 data points to represent 100% of a gait cycle, consistent with standard biomechanical practice (Figure A1). This segmentation approach resulted in nearly 30,000 gait cycles in total. We used this segmentation to determine the best performing sensor configuration to test on different sliding window sizes (Figure 2). The second segmentation strategy employed a sliding window approach in which data were extracted using a fixed window size. Sliding window segmentation is more aligned with data-driven modeling approaches and can be applied when step detection is unavailable or considered unreliable, as models do not depend on gait event detection or gait style. We performed a sensitivity analysis using different window sizes based on indications from the literature that window lengths outside the range of 1–2 s may lead to unreliable representations of gait dynamics (cf. [28,29]).

### 2.3. Machine Learning Approach

#### 2.3.1. Feature Engineering and Reduction

For the machine learning approach, feature engineering was performed prior to model development using the Python 3.10 (Python Software Foundation) packages Pandas [30], Numpy [31], and Scikit-learn [32]. Statistical and frequency-based features were calculated for each gait cycle and sliding window. The statistical features included minimum, maximum, mean, standard deviation, interquartile range, mean absolute deviation, area under the curve, skewness, kurtosis, and entropy using a histogram-based method with ten bins. Frequency-based features were the first five discrete Fourier transform coefficients and the weighted mean frequency. This feature set is similar to those available in packages such as time series feature extraction based on scalable hypothesis tests (TSFRESH) [33]; due to the nature of the feature engineering, our approach inscribes itself into the realm of statistically enhanced learning [34]. Feature reduction was performed using filter methods in order to optimize the dataset (cf. [35]). For features with a correlation threshold of 0.8 or higher, only one feature (randomly selected) from each correlated set was retained. Features with no variation across the dataset and those with a low variation coefficient (less than 0.1) were removed. This approach to feature extraction follows common good practice and should help to define a reproducible method.

#### 2.3.2. Model Training and Evaluation

The preprocessed and reduced dataset was split into training (80%) and test (20%) sets using a subject-wise splitting approach to ensure that all data from a single participant were exclusively in either the training or the test set [20]. The Extreme Gradient Boosting (XGBoost) algorithm [36] was selected as the ML model for classification due to its robustness and high performance in structured data classification tasks. We implemented a stratified group k-fold cross-validation approach to tune the hyperparameters. We ensured that the training and validation folds did not contain data from the same participant, thereby reducing the risk of data leakage and improving the model’s generalizability [19]. Hyperparameters such as the learning rate, maximum depth, and number of estimators were optimized using a grid search approach. We assessed model stability by repeating the training and evaluation steps ten times with different random seeds for the data split. Model performance was evaluated using the mean and standard deviation value of several metrics (accuracy, F1-score, sensitivity, and specificity) across the ten repeated trials. Sensitivity and specificity were included to align with clinical evaluation standards.

### 2.4. Deep Learning Approach

We developed an advanced deep learning pipeline using the same subject-wise data splitting strategy as mentioned earlier. All input signals were standardized using z-score transformation. The final architecture integrated multiple temporal-processing components designed to capture different scales of gait dynamics (Figure A2). The initial residual Conv1D blocks extract robust local temporal–spatial features while improving gradient flow and reducing overfitting. These are followed by a temporal convolutional network (TCN) module with dilated convolutions to capture stride-level and multi-stride dependencies. A bidirectional LSTM layer then models the longer-range sequential structure inherent in cyclic gait patterns. Finally, a multi-head self-attention (MHA) mechanism adaptively weights the most informative time steps, enhancing interpretability by revealing which temporal regions contribute most strongly to classification. To optimize performance, we applied Keras Tuner’s random search to tune key hyperparameters, including the number of convolutional filters, convolutional kernel sizes, LSTM units, number of attention heads, dense layer width, dropout rate, and learning rate. The search space included three learning rates (1 × 10^−4^, 5 × 10^−4^, 1 × 10^−3^) and the selected learning rate remained constant throughout training (i.e., no external learning-rate decay schedule was used). All models were trained using the Adam optimizer with a batch size of 32, categorical cross-entropy loss, and class weights computed from training-set label frequencies to compensate for class imbalance. Training employed an early-stopping rule monitoring validation loss, with a patience of five epochs, min_delta = 0, and automatic restoration of the best-performing weights. Each search trial was trained for up to ten epochs, and the final selected model was retrained for up to fifty epochs under the same early-stopping criterion. All experiments were executed on a workstation configured with TensorFlow 2.16, CUDA-enabled GPU acceleration (NVIDIA hardware), and a mixed CPU/GPU compute. This type of hybrid CNN–TCN–LSTM–attention architecture is well-supported in the gait literature, as combining convolutional, recurrent, and attention mechanisms has been shown to effectively model both fine-grained and long-range temporal dependencies in gait [37,38,39].

### 2.5. Model Comparison

To test our hypotheses for the ML and DL models, we performed eight separate paired *t*-tests of the F1-scores (α=0.05). For our first hypothesis, we tested the difference in model performance between the IMU feet and pressure feet models and the IMU pelvis and pressure model. To test whether adding pressure insoles to IMUs enhances model performance, we tested the difference between the minimal IMU only model, and the combined IMU (feet or pelvis) + pressure feet model. This resulted in a candidate (i.e., best-performing) model for the ML and DL approaches. These models were trained and tested on biomechanical gait segmentation. For our second hypothesis, we tested the difference between segmentation approaches. First, we compared the 1-s and 1.5-s sliding window to the benchmark 2-s sliding window. Second, we used the best sliding window configuration for testing against the candidate model using gait segmentation. If no differences were found between the window sizes, we used the 2-s window. As we also wanted to know whether our minimally-intrusive model performed better than a state-of-the-art full IMU sensor model (established model); thus, we tested the difference between the candidate model and the IMU lower-limbs model (see Figure 2 for the flowchart of model learning). No statistical corrections for multiple comparison were performed.

## 3. Results

Table 1 and Table 2 summarize the results of our candidate model identification and the sliding window analysis performed on that model.

### 3.1. Sensor Input

For the ML models, configurations using IMU feet (F1=0.71) significantly outperformed those using pressure feet (F1=0.66) (t=3.68, p<0.01). Combining modalities (IMU feet + pressure feet) (F1=0.80) significantly improved classification performance over IMU feet alone (F1=0.71) (t=−9.644, p<0.001). Similarly, the IMU pelvis configuration (F1=0.74) showed significantly better classification accuracy over pressure insoles (F1=0.66) (t=4.32, p<0.01), with the combination (F1=0.82) performing even better (t=−5.396, p<0.001). Our candidate (IMU pelvis + pressure feet) model (F1=0.82) was not able to reach the performance of the established IMU lower-limbs model (F1=0.87) (t=−5.19, p<0.01). The confusion matrix for the ML candidate model is shown in Figure 3a.

For the DL approach, the IMU feet models (F1=0.8) significantly outperformed the pressure feet models (F1=0.66) (t=6.16, p<0.001). However, combining IMU feet with pressure insoles did not significantly improve classification over the IMU feet model (F1=0.80 and F1=0.80, respectively) (t=−0.048, p=0.96). In parallel, the pelvis IMU model (F1=0.80) produced significantly better classification accuracy over pressure insoles (F1=0.66) (t=4.11, p<0.01); combining the modalities (F1=0.82) did not lead to any significant improvements (t=−1.69, p=0.12). Similar to the machine learning model, our candidate (IMU pelvis + pressure feet) model (F1=0.82) did not perform as well as the established IMU lower-limbs model (F1=0.89) (t=−2.56, p=0.03).

### 3.2. Segmentation

In the ML model, segmentation technique significantly influenced model performance; the 1-s segmentation window (F1=0.79) improved performance by two percentage points over the 2-s segmentation window (F1=0.77) (t=3.28, p=0.01). Nevertheless, gait segmentation (F1=0.82) outperformed the sliding window (F1=0.79) technique for the combined IMU pelvis and pressure feet model (t=3.17, p=0.01).

In the DL model segmentation technique did not influence classification performance; the 1-s and 1.5-s windows did not significantly differ from the 2-s sliding window. Moreover, the slight increase in F1-score for the 2-s sliding window (F1=0.83) was not significantly different from the gait segmentation (F1=0.82) in our candidate model (t=−0.79, p=0.451). The confusion matrix for the IMU feet + pressure feet model preprocessed using the 2-s sliding window segmentation approach is shown in Figure 3b. As a reference, the established model (IMU lower-limbs model) using the gait segmentation approach is shown in Figure 3c.

## 4. Discussion

### 4.1. Summary

This study investigated the performance of different sensor configurations and segmentation strategies in classifying walking conditions using ML and DL models. Across sensor configurations, models using IMU data consistently outperformed those using pressure insoles alone, reaffirming the superior discriminative capability of IMU sensors for walking condition classification. Combining IMUs on the foot or pelvis with pressure insole data produced results comparable to IMU foot-only or IMU pelvis-only configurations for DL models; on the other hand, it significantly improved surface classification for ML models. Machine learning significantly benefited from gait segmentation over sliding window segmentation. Conversely, there were no distinct differences between the two segmentation methods implemented for deep learning. Thus, the data-driven sliding window approach may be preferable for deep learning, since it does not rely on gait event detection algorithms.

### 4.2. Effect of Sensor Input

The findings partially support our first hypothesis, confirming that IMU sensors provide more discriminative information than pressure insoles across both modeling approaches, at least for the technology (resistive, eight fields), particular model, and normalized pressure data used. The highest F1-scores were found for the established IMU lower-limb models, reaching F1=0.87 in the ML approach and F1=0.89 in the DL approach. In contrast, the models relying solely on pressure insole data demonstrated substantially lower performance, with F1-scores of 0.66 for the ML and DL models. ML models combining both sensor modalities showed gains of approximately nine percentage points in performance compared to the IMU-only models. In contrast, the DL models only showed minimal (non-significant) gains of two percentage points when combining data from the pelvis IMU with pressure insoles. This indicates that although pressure insoles capture complementary aspects of foot–ground interaction, their integration with IMU data only provides a benefit when using ML. Finally, the combined model (IMU pelvis + pressure feet) was not able to achieve comparable performance to the established IMU lower-limbs model, reaffirming that additional information from sensors on the lower limbs is important in classifying surfaces during walking.

### 4.3. Effect of Trial Segmentation Method

Our second hypothesis is supported for ML, but not for DL. ML model performance was significantly influenced by sliding window length, with a length of 1 s proving more beneficial compared to 2 s. Moreover, gait cycle segmentation (F1=0.82) significantly outperformed 1-s sliding window segmentation (F1=0.79) when using the combined IMU pelvis + pressure feet model. For DL models, inspection revealed a difference of one percentage point in favor of sliding window (F1=0.83) over gait cycle segmentation (F1=0.82), although the difference was not significant. Similar classification accuracy with the sliding window approach greatly facilitates data processing and can make DL models more robust, since no step-detection algorithm needs to be implemented. Moreover, as DL performance was not affected by window size, it is preferable to use a 2-s window. This reduces the total number of segments, thereby lowering computational cost and inference time. Approximately 30,000 segments were obtained using this configuration, which is a comparable volume of data to the gait cycle segmentation approach.

In the current study, we used ground-truth annotated steps present in the pressure dataset for the gait segmentation approach. We know from biomechanical work that inaccuracies in initial contact events influence gait joint kinematics calculations [40]. Detecting initial contact can be challenging, as it depends on the method used and surface incline, with errors up to 5.3 ms [41]. As such, we performed a perturbation test of the initial contacts (up to 20 ms). For the ML approach, the results show that accurate initial contact detection is needed, as model performance with perturbed gait cycles decreased by two percentage points from unperturbed gait cycles (Appendix A, Table A2). For the DL approach, given the similar performance between gait and sliding window segmentation, it is unsurprising that perturbation of gait events had no effect (Appendix A, Table A2).

### 4.4. Comparison to Previous Work

Compared to previous work, our results show an improvement in the classification of outdoor walking conditions on several fronts. Because it is not always feasible to obtain surface calibration per participant, it is advantageous in terms of deployment capabilities to develop models that do not require this additional processing step. In our data processing, we used subject-wise stratification to ensure that no data from the same participant were present in both the training and test sets. Of the seventeen studies we found that classified surface topology, only four used a subject-wise or leave-one-subject-out (LOSO) training–test split [10,14,22,42] (Table A1). Of these, Ng et al. [22] found an area under the curve of 0.80 using an SVM model with a single sensor on the right ankle; however, they trained only a binary classifier capable of predicting irregular vs. regular surfaces. More comparable to our work are the studies of Shah et al. [10] and Kobayashi et al. [42]. Kobayashi et al. [42] used a smartphone (location unspecified) to capture six different surfaces; however, they were only able to achieve an accuracy of 44.9% using a LOSO split. In order for their model to perform well, subject data needed to be captured and used in training and testing (accuracy increased to 83.5%). Shah et al. [10] achieved a classification F1-score of 0.78 using a subject-wise split classifying nine outdoor surfaces with either the IMU lower-limbs model or a model using only an IMU on the right shank. Our combined model achieved slightly more favorable F1-scores (DL sliding window, 0.83). Moreover, the outdoor walking in this work is unconstrained, meaning that participants walked a continuous course while encountering different surfaces without stopping. As such, our minimally intrusive IMU feet + pressure feet sensor set-up shows improved results compared to previous work, with the benefit of including a sampling of more clinically relevant surfaces and being recorded in a context that more closely reflects real-life walking conditions (improved ecological validity).

### 4.5. Surface Specific Performance

We found the best ML model to be excellent at classifying stair negotiation (accuracy of 0.94 and 0.93 for stairs down and stairs up, respectively); however, it struggled with slopes (accuracy of 0.59 and 0.88 for slope down and slope up, respectively) (Figure 3a). Slope down was often confused with flat walking 40%. This may be explained by the sloped surfaces used in this study; contrary to most surface classification papers, our dataset contains two slope types, one that is long (50–70 m) with a gradient of 6% and another that is short (3 m) with a gradient of 15%. Moreover, across the dataset, there are fewer instances of steep slope compared to shallow slope (60 vs. 300, respectively). Comparatively, the Luo et al. [43] dataset featured in [10,11,12,13] has a slope of 8.33%, while Chen et al. [44] studied a slope of 36%. A slope with a gradient of 6% is also quite shallow (less than the maximum of 8.33% requirement for Americans with Disabilities Act (ADA)-compliant ramps) and presumably not overly challenging or requiring major biomechanical adaptations for the study population, making it difficult for the model to distinguish from flat walking. On the contrary, the DL model is better able to predict slope down (accuracy of 0.86), but struggles with classifying slope up (accuracy of 0.50). The model incorrectly identified 50% of slope down trials as level walking, compared to 14% of slope down trials. The ecological validity of data from this dataset, including the different slopes and the changes in direction during flat walking, makes our results more generalizable in classifying surfaces in real life conditions.

### 4.6. Limitations and Future Work

There are four main limitations of this study that warrant discussion and can be used to guide future work; these are related to features, model architectures, sample populations, and sensor set-up.

First, in the ML model we relied on generic (statistical- and frequency-based) features instead of domain specific-features. This may have hindered the model’s ability to fully leverage the available gait information. This was done intentionally, as we sought a fair comparison with the DL model. Incorporating (biomechanical) domain-specific features may provide models with more discriminative information and improve their predictive performance. In addition, we did not leverage advanced feature selection techniques (e.g., Salp Swarm Algorithm [45]) or feature extraction techniques that incorporate sensor fusion (e.g., Time Series Fusion (TSFuse) [46]). The Salp Swarm Algorithm, as used in Chauhan et al. [11], effectively reduces feature dimensionality while maintaining high accuracy. It is unclear whether Chauchan et al. [11] used a subject-wise split, making direct comparison with our results difficult. Moreover, we considered all our input data as univariate, assuming no interaction between the signals. Because we used a multiple-sensor set-up, we might have missed information related to interactions between the different sensors. De Brabandere et al. [46] developed an automated feature construction system (TSFuse) that fuses data from multivariate time series both within and between sensors, resulting in the creation of new and possibly relevant time series [46]. Future work could investigate the effect of these advanced and automated feature extraction tools to further refine model performance. Additionally, calculating elevation changes between consecutive gait cycles or sliding windows could help to disambiguate sloped from flat walking, thereby reducing misclassifications.

Second, we only focused on two model architectures: a 1D CNN for the DL approach and XGBoost for the ML approach. The 1D CNN was chosen for its effectiveness in capturing local temporal patterns in sequential sensor data while maintaining a relatively low computational cost compared to more complex architectures such as LSTM Units. Its structure is well-suited for processing multivariate time series such as IMU and pressure insole data, making it an efficient and interpretable choice for our classification task. XGBoost was selected for its strong performance with structured feature-based data, robustness to overfitting, and efficiency in handling high-dimensional inputs. Although we limited our exploration to these two well-established architectures in order to maintain a focused and interpretable evaluation, future work may benefit from exploring alternative or more advanced models tailored to specific sensor modalities.

Third, expanding surface classification studies to different populations remains a very important area for exploration. At this moment, studies have only used asymptomatic and uninjured subjects. While it is important to investigate the capabilities of classifying outdoor walking surfaces in unconstrained individuals, this limits the generalizability of the resulting models. It is unclear whether these models would misclassify surfaces due to an underlying gait pathology or when using walking aids. Therefore, future work needs to include participants with musculoskeletal and/or neurological injuries in order for these models to be applicable for clinical biomechanics in free living environments.

Lastly, our models depend on a complex sensor set-up. Achieving the best prediction F1-scores required seven IMU sensors. Even though we achieved good accuracy (F1 > 0.9) for flat walking and stairs using the IMU feet plus pressure insole models, classifying sloped walking remained challenging. Future work should investigate methods for extracting relevant features capable of recognizing sloped walking, allowing for improved real-world applicability and feasibility.

### 4.7. Code Availability

The custom Python notebooks for preprocessing the data along with the model creations are provided at https://github.com/mcgillmotionlab/SurfaceClassification_FootPressure_IMU (accessed on 20 December 2025).

## Figures and Tables

**Figure 1 sensors-26-00232-f001:**
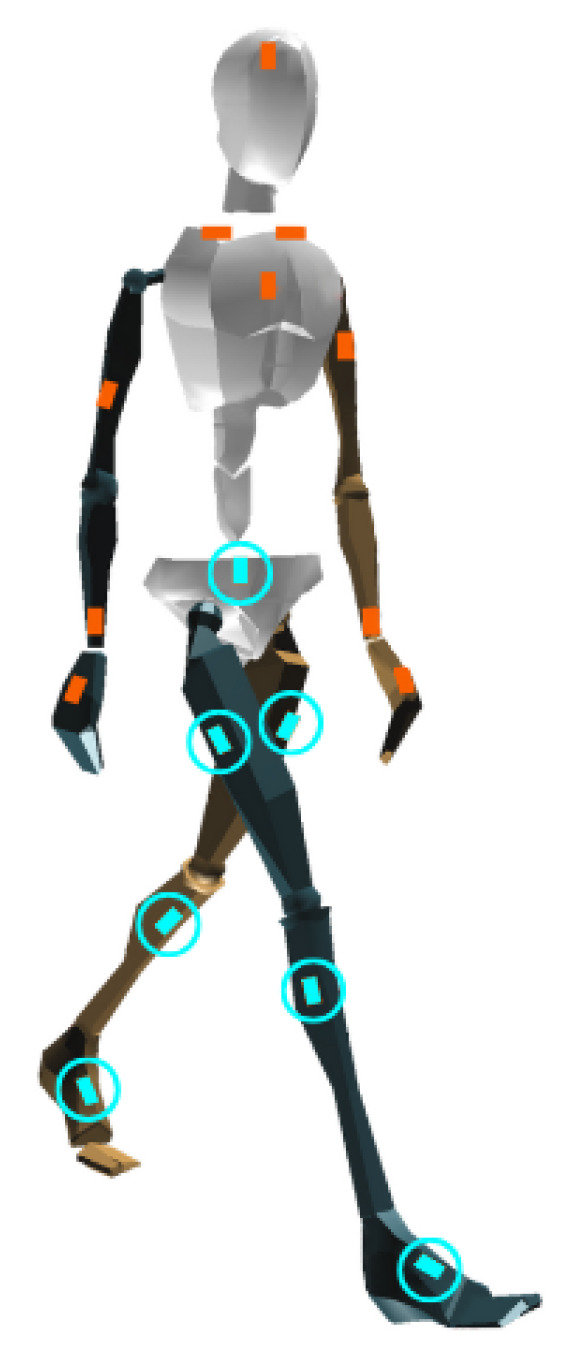
Schematic representation of the IMU sensor set-ups used in this study. The seven-sensor “lower limb configuration” (pelvis, upper legs, lower legs, and feet) includes all of the sensors circled in teal.

**Figure 2 sensors-26-00232-f002:**
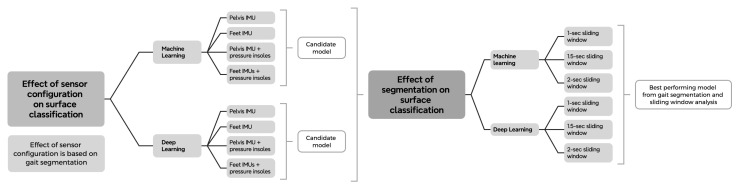
Flow chart of data learning processes within this manuscript.

**Figure 3 sensors-26-00232-f003:**
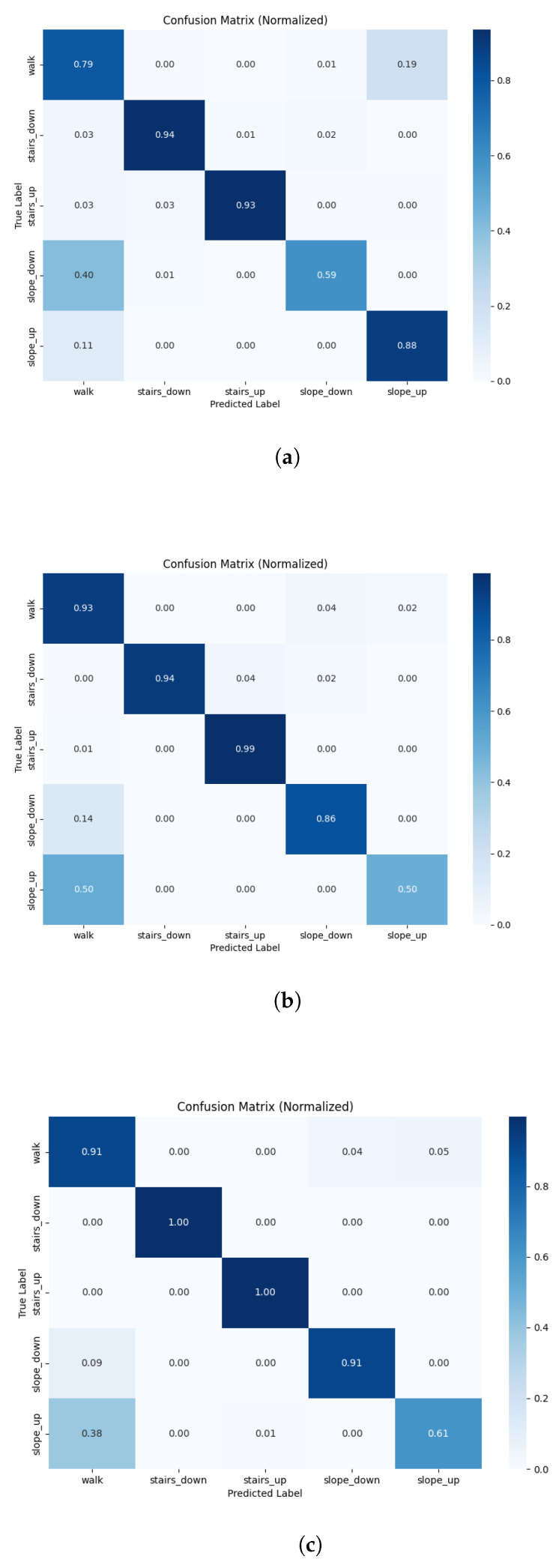
Representative normalized confusion matrices of surface classification models (walk, stairs down, stairs up, slope down, slope up) for (**a**) machine learning IMU pelvis + pressure feet model based on gait cycle segmentation (F1=0.82); (**b**) deep learning IMU pelvis + pressure feet model based on 2-s sliding window segmentation (F1=0.83); and (**c**) deep learning IMU lower-limbs model based on gait cycle segmentation (F1=0.89).

**Table 1 sensors-26-00232-t001:** Performance metrics for different sensor configurations of the machine learning and deep learning models using gait cycle segmentation.

Model	Sensor Configuration	Accuracy	F1-Score	Sensitivity	Specificity
ML	IMU lower-limbs	0.85 (0.05)	0.87 (0.04)	0.86 (0.03)	0.94 (0.01)
	IMU feet ^a^	0.70 (0.05)	0.71 (0.03)	0.71 (0.04)	0.89 (0.01)
	IMU pelvis ^a^	0.74 (0.07)	0.74 (0.05)	0.75 (0.04)	0.92 (0.02)
	Pressure feet	0.69 (0.05)	0.66 (0.03)	0.65 (0.03)	0.89 (0.01)
	IMU feet + Pressure feet ^b^	0.81 (0.04)	0.80 (0.04)	0.79 (0.04)	0.93 (0.01)
	**IMU pelvis + Pressure feet ^b,c^**	**0.81 (0.05)**	**0.82 (0.04)**	**0.81 (0.03)**	**0.93 (0.02)**
DL	IMU lower-limbs	0.89 (0.03)	0.89 (0.04)	0.87 (0.04)	0.96 (0.01)
	IMU feet ^a^	0.78 (0.05)	0.80 (0.04)	0.79 (0.02)	0.92 (0.01)
	IMU pelvis ^a^	0.81 (0.07)	0.80 (0.07)	0.80 (0.06)	0.93 (0.02)
	Pressure Feet	0.73 (0.05)	0.66 (0.06)	0.63 (0.06)	0.90 (0.02)
	IMU feet + Pressure feet	0.80 (0.07)	0.80 (0.05)	0.78 (0.05)	0.93 (0.02)
	**IMU pelvis + Pressure feet ^c^**	**0.85 (0.04)**	**0.82 (0.07)**	**0.81 (0.06)**	**0.94 (0.01)**

Mean (standard deviation) shown. See main text for details on sensor configurations. Best experimental model shown in bold. IMU lower-limbs model shown for comparison purposes. Significant differences are indicated with superscript letters. ^a^: IMU-only model is significantly different from pressure-only model; ^b^: Combined model is significantly different from IMU-only model; ^c^: Best experimental model is significantly different from established lower-limbs model. Abbreviations: Inertial Measurement Unit (IMU).

**Table 2 sensors-26-00232-t002:** Performance metrics for different window lengths of the machine learning and deep learning candidate models.

Model	Window Length	Accuracy	F1-Score	Sensitivity	Specificity
ML	**1-s**	**0.81 (0.04)**	**0.79 (0.04)** ^a^	**0.78 (0.03)**	**0.93 (0.01)**
	1.5-s	0.76 (0.07)	0.76 (0.04)	0.77 (0.03)	0.92 (0.02)
	2-s	0.77 (0.06)	0.77 (0.04)	0.77 (0.03)	0.92 (0.02)
DL	1-s	0.83 (0.05)	0.82 (0.05)	0.81 (0.04)	0.94 (0.01)
	1.5-s	0.84 (0.07)	0.83 (0.08)	0.83 (0.07)	0.94 (0.02)
	2-s	0.84 (0.07)	0.83 (0.08)	0.83 (0.06)	0.94 (0.02)

Model tested is the best experimental model (IMU Pelvis + Pressure feet configuration). See main text for details on sensor configurations. Mean (standard deviation) shown. Significant differences are indicated with superscript letters: ^a^: F1-score for that segment length is significantly different from the 2-s one. Best model shown in bold (only for ML as DL models were not significantly different).

## Data Availability

The data presented in this study were derived from the following resources available in the public domain on figshare: https://doi.org/10.6084/m9.figshare.c.5758997.v1.

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
