# Peer review of "Outdoor Walking Classification Based on Inertial Measurement Unit and Foot Pressure Sensor Data"

_sensors, 2025, doi:10.3390/s26010232_

Round 1
Reviewer 1 Report
Comments and Suggestions for Authors
This study presents a well-executed machine learning and deep learning framework for classifying walking conditions using multi-sensor data (IMU and pressure insoles) from a publicly available dataset. The manuscript is clearly structured, methodologically sound, and the conclusions are supported by data. Nonetheless, several critical aspects require clarification, particularly regarding model robustness, sensor reduction strategies, and clinical applicability.
***Major Concerns***
- The models were trained and validated using data from healthy participants only. Although the authors briefly acknowledge this limitation in section 4.6, they do not quantify or simulate how pathological gait (e.g., due to Parkinson’s disease or stroke) would affect classification performance. Given the motivation of real-world mobility assessment, a more rigorous discussion—ideally backed by sensitivity simulations or adversarial perturbations—should be included.
- The study reports poor performance in distinguishing slope down vs. level walking (especially in ML models, Fig 2a), yet provides only a brief explanation that different gradient slopes (6% and 15%) might be the cause. However, the exact distribution of slope types in the dataset is not provided. A breakdown of performance across slope subtypes is critical to substantiate this claim and would clarify whether model confusion is due to biomechanical similarity or data imbalance.
- The optimal model configuration requires seven IMUs and pressure insoles, which raises practical concerns for real-world deployment. While the authors mention the goal of reducing sensors in future work, no performance trade-off analysis is provided. Including ablation studies (e.g., using only two or three sensors) or referencing wearable-friendly configurations from prior work would substantially improve translational value.
- The pressure data used in the DL model is raw, yet pressure data is known to benefit from engineered features such as center-of-pressure trajectory, stance/swing ratios, and derived gait phases. The manuscript would benefit from a supplemental experiment comparing raw vs. feature-enhanced pressure insole input to demonstrate whether the current implementation underutilizes the modality.
- The study claims that both gait cycle segmentation and sliding window approaches are viable, with a slight performance edge for sliding window in DL. However, there is no quantitative analysis of the gait event detection quality in the dataset or how errors in segmentation affect classification. Given the dependence of gait-based segmentation on accurate foot contact labeling, a discussion on reliability or error propagation is warranted.
- Model metrics (F1-score, accuracy, etc.) are presented as mean± SD across 10 random seeds, but no formal statistical test (e.g., paired t-test or bootstrap confidence intervals) is used to determine whether performance differences between sensor setups or segmentation strategies are significant. This weakens the comparative claims in the discussion.
- It would help readers if sensor orientations or axes (e.g., AP, ML, vertical) were described in relation to anatomical planes, especially given the re-orientation from global to local frames.
- Some phrases such as “IMU sensors are more discriminative” are repeated with minimal elaboration. Consider consolidating similar statements in the Discussion.
- Some references (e.g., [9], [10]) appear frequently but are not critically evaluated. Consider integrating a comparative table or structured discussion highlighting strengths/weaknesses of prior approaches vs. current work.
- Use consistent terminology for walking conditions. Sometimes “flat”, “level”, and “straight path” appear interchangeably—this may confuse readers.
Author Response
This study presents a well-executed machine learning and deep learning framework for classifying walking conditions using multi-sensor data (IMU and pressure insoles) from a publicly available dataset. The manuscript is clearly structured, methodologically sound, and the conclusions are supported by data. Nonetheless, several critical aspects require clarification, particularly regarding model robustness, sensor reduction strategies, and clinical applicability.
***Major Concerns***
1. The models were trained and validated using data from healthy participants only. Although the authors briefly acknowledge this limitation in section 4.6, they do not quantify or simulate how pathological gait (e.g., due to Parkinson’s disease or stroke) would affect classification performance. Given the motivation of real-world mobility assessment, a more rigorous discussion—ideally backed by sensitivity simulations or adversarial perturbations—should be included.
We appreciate that assessment of model performance on patients with specific pathologies would be useful. To date, it is not possible to accurately simulate the pathologies mentioned by the reviewer. Thus, it would be necessary to collect additional data in these populations to gauge the generalizability of our models. As the reviewer mentioned, we highlighted this limitation in the discussion. Future work is planned to explore performance of models in clinical populations.
2. The study reports poor performance in distinguishing slope down vs. level walking (especially in ML models, Fig 2a), yet provides only a brief explanation that different gradient slopes (6% and 15%) might be the cause. However, the exact distribution of slope types in the dataset is not provided. A breakdown of performance across slope subtypes is critical to substantiate this claim and would clarify whether model confusion is due to biomechanical similarity or data imbalance.
Further analysis of the dataset reveals that there are only 60 instances of the steep slope, but 300 instances of the shallow slope. Moreover, as the shallow slopes are much longer than the steep slope (50-70m compared to 3m, respectively), after segmentation, the model learns disproportionally the characteristics of shallow slopes. This may explain the difficulty in segregating slopes from level walking. Unfortunately, as both slope types are labeled identically, it was not feasible to train new models with the 2 different slope conditions. We added details regarding the instances of each slope type in the discussion.
3.The optimal model configuration requires seven IMUs and pressure insoles, which raises practical concerns for real-world deployment. While the authors mention the goal of reducing sensors in future work, no performance trade-off analysis is provided. Including ablation studies (e.g., using only two or three sensors) or referencing wearable-friendly configurations from prior work would substantially improve translational value.
Based on this comment, we trained a new foot-only IMU model. This model does not perform as well as the 7 IMU model (F1 score of 0.65-0.81 compared to 0.87-0.91, respectively), but shows that a wearable-friendly configuration is feasible. Moreover, this new model allows for “fairer” comparison with the pressure insole model (both models use sensors on the feet only). We also include a full summary of existing surface classification models (see table in appendix) and discuss the performance of our models with respect to the state-of-the-art. Note, due to space constraints, a full ablation study was not performed. Moreover, we removed the “Luo” configuration as it is similar to the 7 IMU model.
4. The pressure data used in the DL model is raw, yet pressure data is known to benefit from engineered features such as center-of-pressure trajectory, stance/swing ratios, and derived gait phases. The manuscript would benefit from a supplemental experiment comparing raw vs. feature-enhanced pressure insole input to demonstrate whether the current implementation underutilizes the modality.
We avoided using “domain-knowledge” based features in both the IMU and foot pressure models to allow for fair comparison. Nonetheless, we appreciate that some pre-processing would be fair for the pressure insoles, as simple transformations were also done for the IMUs (e.g. coordinate system transformation). In the revised manuscript, we used normalized pressures (amplitude normalized to the mass of each participant) as inputs to the models. This change in isolation improved F1-scores by 2-3%. Note also that the majority of features described by the reviewer could also be extracted from the IMUs, so we chose not to include them in the pressure models as not to provide an unfair advantage.
5. The study claims that both gait cycle segmentation and sliding window approaches are viable, with a slight performance edge for sliding window in DL. However, there is no quantitative analysis of the gait event detection quality in the dataset or how errors in segmentation affect classification. Given the dependence of gait-based segmentation on accurate foot contact labeling, a discussion on reliability or error propagation is warranted.
Indeed, the reviewer raises an important aspect of our work that we failed to report in the original submission. In the revised manuscript (supplemental figure 1), we plot the sagittal plane knee angles, normalized to 100% of the gait cycle, available from the processed IMUs to gauge the success of the gait segmentation strategy. This particular output was chosen as the time-series have a shape easily recognizable by biomechanists working in gait. While we did not quantitatively analyze the results, the plots show that very few cycles were incorrectly portioned during the gait segmentation process. We believe a discussion on reliability or error propagation is outside the scope of the current work. We reference this figure in the updated methods section 2.2.2
6. Model metrics (F1-score, accuracy, etc.) are presented as mean± SD across 10 random seeds, but no formal statistical test (e.g., paired t-test or bootstrap confidence intervals) is used to determine whether performance differences between sensor setups or segmentation strategies are significant. This weakens the comparative claims in the discussion.
Indeed, it was difficult to make comparative claims without a formal analysis. We have now included paired t-tests to support any claims of difference. The analysis approach is detailed in section 2.5 and the results section is updated with related p-values and other statistics.
7. It would help readers if sensor orientations or axes (e.g., AP, ML, vertical) were described in relation to anatomical planes, especially given the re-orientation from global to local frames.
Indeed, we agree with the reviewer that the coordinate system of the sensors may have been unclear in the original submission. In the revised text (section 2.2), we state that the sensors are all transformed in line with the recommendations of the International Society of biomechanics, with new reference 25.
8. Some phrases such as “IMU sensors are more discriminative” are repeated with minimal elaboration. Consider consolidating similar statements in the Discussion.
Based on this comment and the previous one, we now provide details of our formal statistical analysis when making these comments in the discussion.
9. Some references (e.g., [9], [10]) appear frequently but are not critically evaluated. Consider integrating a comparative table or structured discussion highlighting strengths/weaknesses of prior approaches vs. current work.
Based on this comment and comment 3 from the same reviewer, we have now included a summary table (supplemental Table 1) of 17 papers describing existing surface classification models. Relevant papers are also further described in the introduction, allowing us to make a stronger case for the aims of this study. From the review of the literature, it is clear that performance of foot pressure is not well established.
10. Use consistent terminology for walking conditions. Sometimes “flat”, “level”, and “straight path” appear interchangeably—this may confuse readers.
We thank the reviewer for noticing this inconsistency. The language has been unified in the revised manuscript. We hope this will clarify the manuscript for the reader. All mentions of level have been replaced by flat.

Reviewer 2 Report
Comments and Suggestions for Authors
The study presents a relevant and well-structured exploration of wearable-sensor–based walking condition classification. However, several methodological and analytical aspects require clarification or expansion to strengthen the scientific rigour and interpretability of the work. The authors are kindly asked to address the following points during revision.
1) The manuscript would benefit from a clearer articulation of its primary novelty relative to prior IMU-based surface classification studies. Could the authors specify more explicitly what differentiates this work from earlier studies using similar datasets and modelling strategies?
2) The confusion matrices reveal substantial misclassification for particular surfaces (e.g., slope-down predicted as level walking in the ML model, and level or stairs-down predicted as slope-up in the DL model). Could the authors provide deeper analysis or interpretation of these error patterns, possibly including surface-specific characteristics, subject-level contributions, or feature-saliency/importance assessments?
3) The rationale behind selecting a 2-second sliding window with a 1-second stride is not fully justified. Could the authors elaborate on the basis for these choices and indicate whether alternative window sizes were tested or might influence classification performance?
4) The pressure insole-only models display notably poor performance relative to IMU-based configurations. Could the authors provide additional explanation regarding whether this limitation is attributable to the insole hardware (e.g., limited sensor resolution), the use of raw pressure signals without derived biomechanical parameters, or other data-quality considerations?
5) Many observed performance differences across sensor configurations and segmentation methods fall within one standard deviation. Have the authors conducted any statistical comparisons across repeated runs (e.g., paired tests or confidence interval analyses) to determine whether these differences are statistically meaningful?
6) The deep learning approach is restricted to a 1D CNN architecture. Could the authors clarify why alternative time-series architectures commonly used in gait analysis, such as LSTM, GRU, or hybrid CNN–recurrent models, were not evaluated or benchmarked?
7) Given the ecological variability in the dataset (e.g., two different slope gradients, changes in direction, heterogeneous outdoor terrain), could the authors clarify whether specific surfaces or course segments contributed disproportionately to misclassification and whether performance differed between the 6% and 15% slope conditions?
Author Response
please see attachement

Round 2
Reviewer 1 Report
Comments and Suggestions for Authors
This revised manuscript presents a machine learning and deep learning framework for classifying walking conditions using IMU and foot-pressure sensor data. The authors have implemented several modifications in response to earlier reviewer feedback, including: (1) introducing a foot-only IMU model, (2) normalizing pressure data, (3) providing statistical testing, (4) adding a literature summary table, and (5) including gait segmentation visualizations. Overall, the revisions improve the clarity and methodological transparency of the work. Nevertheless, a few newly introduced issues warrant further attention.
*** New Issues Identified After Revision ***
- The revised manuscript still evaluates model performance only in clean, controlled conditions. Without perturbation tests (timing distortions, noise injection, sensor dropout), the model’s resilience in uncontrolled outdoor or clinical settings is unknown. This is a major gap for translational applicability.
- The authors’ qualitative knee-angle overlays are insufficient. Segmentation errors directly impair classifier performance; therefore, even minimal quantitative assessment (e.g., cycle-to-cycle phase variance, template RMSE) is essential. The lack of quantitative reliability analysis weakens conclusions about segmentation viability.
- A single foot-only IMU configuration does not constitute a meaningful sensor-reduction study. For real-world wearability, users often tolerate only: (1) 2–3 IMUs total; or (2) a single lumbar IMU; or (3) foot-pressure only.
- Although normalization improved performance by 2–3%, key questions remain: (1)How does normalization affect within-subject vs. between-subject variability? (2) Does normalization change class separability? (3) Is normalization applied per step, per trial, or per participant globally?
- The revised manuscript still lacks detail regarding: (1) learning rate schedules; (2) batch sizes per model; (3) early stopping rules; (4) GPU/CPU computational setup; (5) seed-matching across model comparisons.
- The authors continue to use a fixed 2-second or similar window without reporting: (1) sensitivity to window size; (2) whether shorter windows degrade accuracy; (3) whether larger windows unnecessarily increase latency.
- The revised manuscript includes 7 self-citations written by the corresponding author. After reviewing their placement and relevance, my assessment is: Instances where the papers are cited to support routine statements (e.g., IMU advantages, generic gait analysis assumptions, or trivial methodological framing). These could be replaced by more standard literature and may give the impression of citation padding. Approximately half of the self-citations appear justified, while several are optional and not essential to the scientific argument. The authors should review each citation and ensure that only methodologically indispensable references remain.
Reviewer 2 Report
Comments and Suggestions for Authors
I am willing to accept the paper in its current form.
Author Response
Thank you for your review.